**communications** engineering

# Aerodynamics-guided machine learning for design optimization of electric vehicles
Jonathan Tran [1], Kai Fukami [1], Kenta Inada[2], Daisuke Umehara[2], Yoshimichi Ono[2], Kenta Ogawa[2] & Kunihiko Taira [1] ✉

The transition to electric vehicles is driving a fundamental shift in the automobile design process. Changes in constraints afforded by the absence of a combustion engine create new opportunities for modifying vehicle geometries. Current approaches to optimizing vehicle aerodynamics require a vast amount of computational studies and physical experiments, which are expensive when performing parameter sweeps over conceivable geometric configurations, suggesting the need for more efficient surrogate models to assist analysis. Here we analyze a dataset of industry-quality automobile geometries with their associated aerodynamic performance obtained from experimentally validated, high-fidelity large-eddy simulations. We show that a relationship between these geometries and their respective aerodynamics can be extracted in a low-dimensional manner by leveraging a nonlinear autoencoder which is simultaneously trained to estimate the drag coefficient from the latent variables. We perform aerodynamic design optimization of vehicle designs by making use of the learned aerodynamic relationship in the low-order space obtained by the model. We demonstrate that the aerodynamic trends for the geometries produced from the optimization process show agreement with validation simulations. The findings of this work demonstrate the application of data-driven approaches to the analysis and design of vehicles in a production environment.

The shift from traditional combustion-based transportation to electric vehicles has brought about notable changes in vehicle design, particularly in terms of aerodynamics. The absence of a combustion engine has allowed for a reevaluation of design constraints and priorities, offering automotive designers greater creative freedom. For example, the elimination of the requirement for engine cooling has allowed for the transformation of the front geometry of electric vehicles from both a functional and stylistic perspective. However, the transition to electric vehicles also brings new concerns. Even as battery and motor efficiencies improve, the current performance of many electric vehicles struggles to meet the demands of consumers, with many electric vehicles operating with limited driving ranges despite long charging times[1]. The importance of vehicle efficiency has been exacerbated by the transition to sustainable transportation. The departure from conventional design norms is emblematic of the need for novel methods to accelerate the design and product development process of vehicles.

On a broader scope, the transportation sector plays a pivotal role in the global energy landscape. Transportation of people and goods constitutes approximately one-quarter of the world's energy consumption by end users, contributing to nearly 40% of global emissions[2]. Oil accounts for 90% of energy consumption in transportation, with oil for use in road transport alone responsible for almost 45% of global oil demand[2,3]. Additionally, transportation is directly responsible for approximately 14% of total global greenhouse gas emissions producing around 7 billion tonnes of $CO_2$[4,5]. Unfortunately, the environmental burden associated with the transportation sector is only expected to increase. Already worldwide, there are more than 1.4 billion passenger and commercial automobiles—the majority of which are powered by internal combustion engines—with this number only projected to go up[6]. Due to global growth in population and income, the demand for passenger and freight travel is projected to increase up to 41% from 2022 to 2050, which only serves to further agitate the growing climate crisis[7,8]. All of the aforementioned facts suggest that it is especially important for vehicle manufacturers to explore methods to push the boundaries of vehicle efficiency.

Transportation fundamentally relies on the conversion of chemical energy from fuel into mechanical motion. Accordingly, vehicles must waste excess energy to overcome driving resistance. This wasted energy manifests itself as a decrease in fuel efficiency, resulting in increased emissions and reduced range. A notable source of resistance comes from aerodynamic drag, which can account for about 75–80% of the total driving resistance

[1]Department of Mechanical and Aerospace Engineering, University of California, Los Angeles, CA, USA. [2]BEV Automobile Development Unit, Honda Motor Co., Ltd., Tochigi, Japan. ✉e-mail: ktaira@seas.ucla.edu

when driving at 100 km/h[9,10]. In the case of automobiles, a 20% reduction of aerodynamic drag can result in a noticeable 4% decrease in fuel consumption of highway operation[11]. In the case of electric vehicles, without the associated losses due to combustion, the increased efficiency of the powertrain for electric vehicles results in a greater proportion of energy loss due to aerodynamic drag[12,13]. This makes aerodynamic performance a major design priority due to the direct relationship between drag and vehicle efficiency.

As even small geometric changes may have large effects on the flow field around a vehicle—thereby influencing its aerodynamic performance—a careful analysis of the effect of parameter changes on aerodynamics is critical for vehicular design. Large-scale modifications can alter characteristics of the pressure distribution in the flow around a vehicle, which could considerably alter the pressure drag, which can account for up to almost 85% of the total drag on the vehicle[14–16]. Additionally, small-scale changes can modify the boundary layer which can alter drag resulting from friction at the surface. However, predicting the aerodynamic performance is difficult due to the nonlinear nature of fluid flows, with even small uncertainties in estimating drag costing the US billions per year[17].

Aerodynamic analysis of vehicle performance in relevant operating conditions often requires a comprehensive campaign of experimental or computational trials. Experimental studies, including wind-tunnel tests, require a physical car model as well as expensive testing facilities. High-fidelity computational fluid dynamic (CFD) studies based on direct numerical simulation or large-eddy simulations (LES) require substantial computational resources[13]. To resolve the smallest turbulent scales of the flow, the number of grid points scales with the Reynolds number $Re$, a value that measures the ratio between inertial and viscous forces, on the order of $Re^{9/4}$. In the case of turbulent flow past an automobile in which the Reynolds number can well exceed millions, this task quickly becomes very expensive[18].

To make matters more difficult, both numerical and experimental trials need to be performed spanning an extremely large parameter space of vehicle designs with practically infinitely many possible configurations, becoming expensive campaigns in terms of both finance and time. Furthermore, the data size associated with vehicle analyses increases as experimental and computational techniques continuously improve in fidelity and accuracy. Consequently, such large amounts of high-dimensional data can make analysis extremely difficult. Along with such expensive analyses, optimization of vehicle aerodynamic performance also necessitates an extensive amount of manual modeling. These issues naturally call for data-driven approaches, such as machine learning, to learn the underlying physical relationships between vehicle geometries and aerodynamic performance to direct design analysis toward specific cases that are expected to improve aerodynamics. While trained aerodynamicists may be able to manually intuit a relationship between aerodynamic performance and geometric changes, we anticipate that deep machine-learning methods can learn such aerodynamic relationships from a large dataset while offering physical insights into vehicle design, beyond that of human intuition.

The optimization of industrial designs necessitates in-depth, high-fidelity, analysis of complex phenomena and complex geometries, which poses a critical bottleneck to the design process. The iterative nature of such design demands careful consideration of numerous factors and constraints, and any modifications to the design require thorough analysis. In this sense, developing a surrogate model that estimates design parameters of interest that are associated with expensive, high-fidelity, analyses can greatly improve the efficiency of the feasibility study cycle[19]. Performing data-driven shape optimization can help mitigate the cost burden associated with manual modeling and analysis for a naïve parameter sweep by producing modifications that are expected to improve aerodynamic performance, which can then be further studied. This problem setting is depicted in Fig. 1 for the case of automobile geometries.

We perform a data-driven analysis on industry-quality vehicle geometries for the purpose of drag reduction. Given that many high-dimensional data can be explained with fewer coordinates in a low-order manner[20–23], we seek to learn a relationship in an appropriate form for this purpose. Previous studies have demonstrated that a low-order description of vehicle geometries can be leveraged for vehicle shape optimization[24–27]. However, such studies have been limited to relatively lower-fidelity, steady-state, Reynolds Averaged Navier-Stokes (RANS) simulations typically for simplified geometries and simulation conditions. In this work, we seek to utilize a data-driven study of vehicular aerodynamics in an industrial setting. The present analysis is performed on production designs, with aerodynamic performance obtained from high-fidelity Large-Eddy Simulations (LES) utilizing an experimentally validated computational setup[13].

The current data-driven approach as outlined in Fig. 1 is proposed as a method to direct traditional high-fidelity analysis in a cost-effective manner to accelerate the iterative product design process. By employing a modified nonlinear autoencoder, we learn a low-order manifold, which simultaneously contains a nonlinear relationship between vehicle geometries and their associated drag coefficients. We find that incorporating the drag coefficients in the autoencoder training process provides low-order coordinates that are suitable for design optimization. The learned manifold, which captures a nonlinear relationship between vehicle geometry and aerodynamic drag, is leveraged to obtain modifications for a given vehicle geometry to improve aerodynamic performance. The present work demonstrates that this simple data-driven approach sufficiently captures trends in aerodynamically relevant features, which we validate through a CFD analysis of decoded geometries. While the present work primarily focuses on aerodynamic shape optimization, the framework we employ to perform this shape optimization is not specific to the choice of input geometry representations and design parameters.

## Results
### Automobile aerodynamic analysis
Our dataset spans a wide range of industry-quality automobile geometries, which exhibit vastly different unsteady wake behavior and aerodynamic performance, as shown by the voxelized geometries in Fig. 2a. The baseline car designs considered in this study come from different production model car geometries including SUVs, hatchbacks, sedans, and box cars, with multiple vehicle models for each car type considered. Additionally, the features of the baseline designs for each model are parametrically modified to produce a variety of geometries. The flow is computed with a large-eddy simulation utilizing a moving mesh that has been validated with industry wind-tunnel experiments. Further details of the aerodynamic analysis are given in the Methods section.

The time-averaged flow around automobiles shares several salient features that are useful in estimating the aerodynamic drag[14,28–31]. For even the most simplified automobile geometries, three-dimensional separated flow regions are generated in the wake behind the vehicle. The flow past most automobiles exhibits two recirculation regions that are formed when the flow separates at the top and bottom edges of the back of the vehicle. The size of the recirculation regions depends on how the shape of the rear geometry directs the trailing wake flow. A pair of trailing vortices are formed when the fluid boundary layer rolls around the sides of the vertical supports at the rear, referred to as the "C-pillars"[14–16]. These wake structures create a region of low pressure behind the vehicle and contribute to the induced drag. Conversely, a region of high pressure exists at the front of the vehicle where the flow is stopped by the front geometry. The difference in pressure over the surface of the vehicle results in the pressure drag, which is the primary source of drag force for automobiles[14]. Figure 2b depicts the total pressure fields for the different car types in the dataset. The zero total pressure isocontour is shown ($C_{p,t} = 0$), which highlights where the flow may separate from the vehicle. We observe that in addition to the rear wake, we also see flow separation at other regions, including the wheels, the front roof edge, the front windshield edges ("A-pillar"), and the front bumper edges. We note that the use of the moving mesh LES solver captures physical phenomena not seen in previous analyses. One example is the wake generated from the outflow from the rotating front wheel, which produces appreciable differences in the drag estimate but is not captured with a static mesh[13].

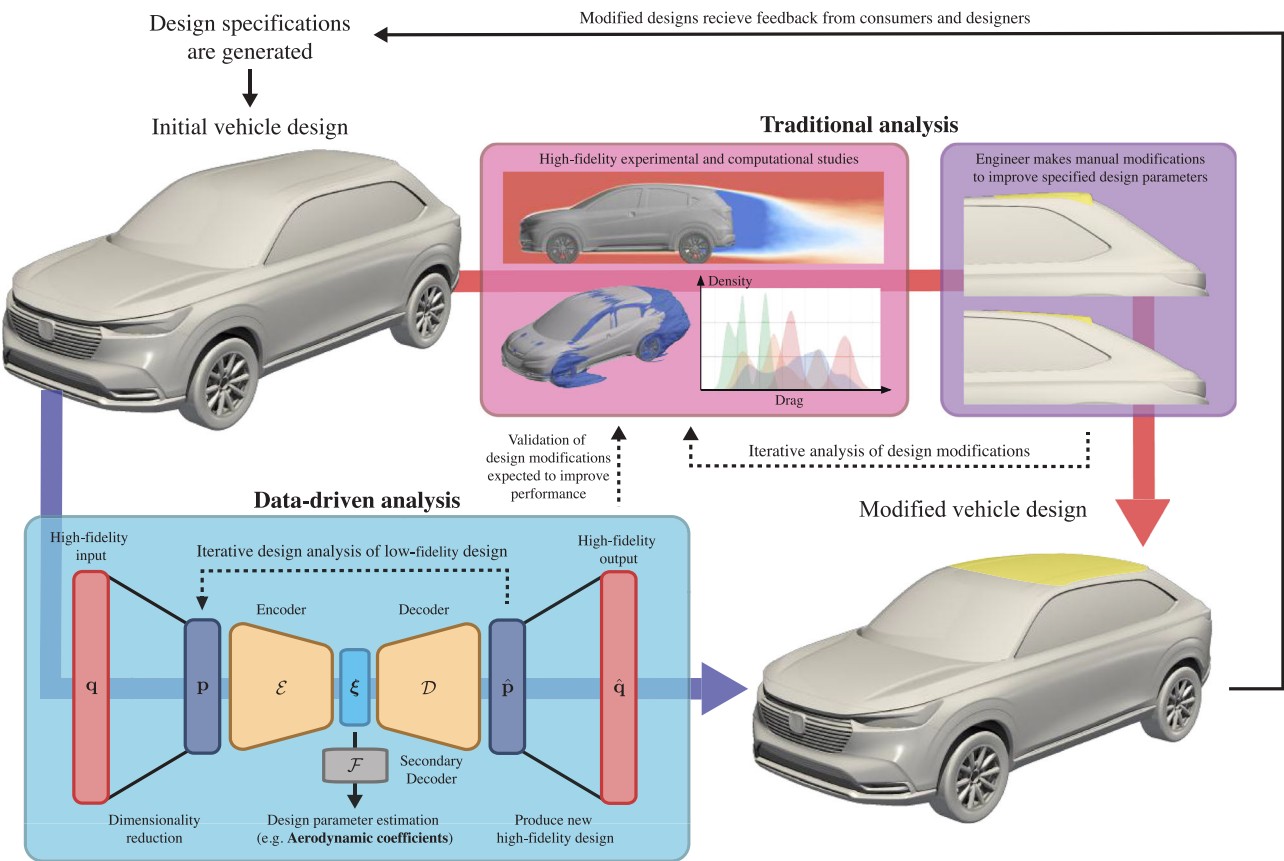

**Fig. 1 | Overview of data-driven directed design analysis.**

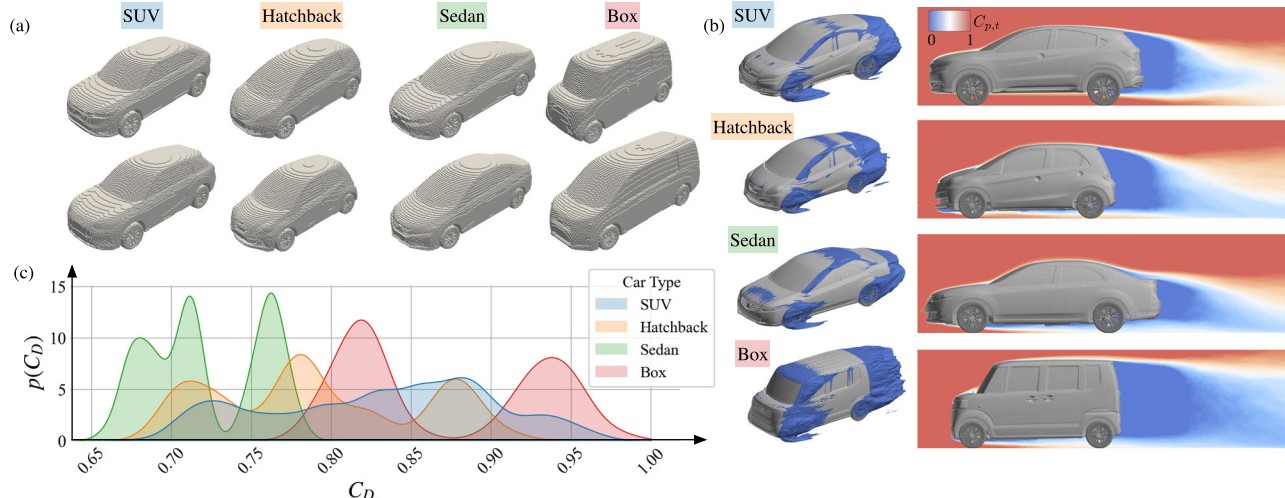

**Fig. 2 | Depiction of the vehicle dataset consisting of sport utility vehicles (SUV), hatchbacks, sedans, and box geometries. a** Representative voxelized vehicle geometries for different car types. **b** Example 3D and 2D (center plane) isocontours of normalized total pressure coefficient, $C_{p,t}$, for different car types. $C_{p,t} = 0$ isocontour is shown, indicating separated flow. **c** Probability density plot, $p(C_D)$, of normalized drag coefficient, $C_D$, with respect to vehicle type.

With the variation of geometric features, the relative sizes of the aforementioned structures and the resulting aerodynamic performance can vary drastically from case to case. For example, if the transition between the front windshield and the top roof panel is abrupt, the flow may separate from the front of the roof (Box). A different flow pattern can appear in the rear where, depending on the rear windshield angle, the flow may separate over the rear spoiler only to reattach lower on the bottom of the windshield (Sedan). Figure 2c shows a probability density plot of the normalized drag coefficients $C_D$, which demonstrates that the different geometries in our dataset span vastly different ranges of drag coefficients.

Although we can qualitatively analyze how fluid structures and aerodynamic performance relate to the vehicle geometry, the wide range of vehicle parameters and the nonlinear nature of fluid flow make it challenging to formulate aerodynamic prediction models, especially when comparing visually similar geometries. For example, the box car exhibits much

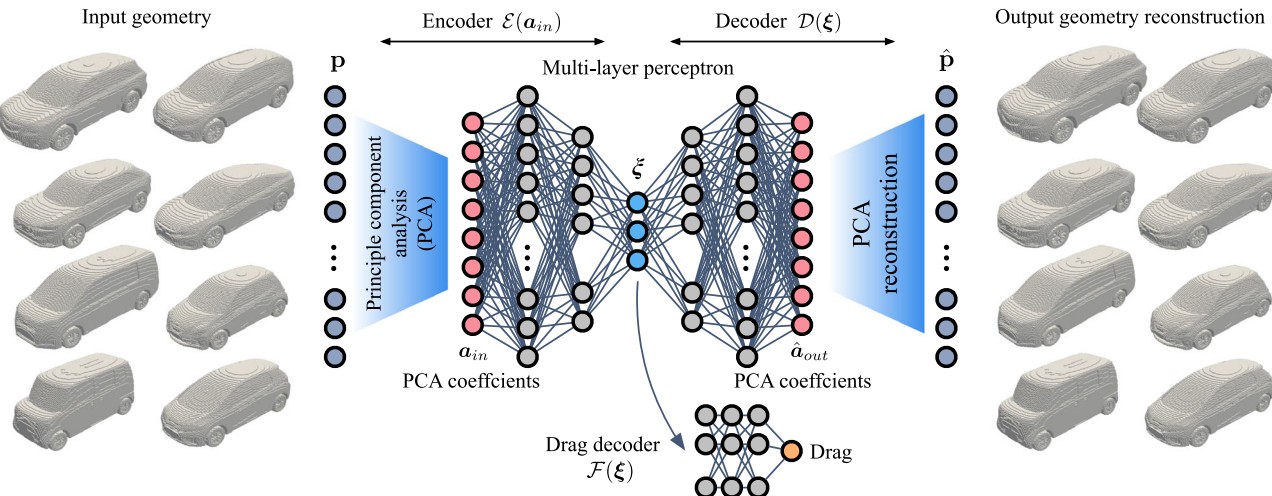

**Fig. 3 | Principal component analysis (PCA) assisted observable-augmented autoencoder.** Voxelized vehicle geometries are taken as input and output, and the drag coefficient is estimated from the encoded latent representation.

larger separated flow regions at the front geometry, as well as a larger low-pressure wake behind the vehicle compared to the other geometries. However, we can see that in Fig. 2c some of the box cars in the current dataset can exhibit a lower drag coefficient than some SUV and hatchback designs, possibly due to differences in the front geometry. Other complexities arise in other flow interactions, such as ground effects in the wake and underbody regions, which can be profoundly altered by minor geometric changes, and require careful aerodynamic analysis. As such, the nonlinear behavior of the flow confounded with a large number of geometric parameters makes a reasonable prediction of the drag performance from geometries difficult to infer. However, we expect that there exists some nonlinear functional dependency between the vehicle geometry and its drag coefficient that can be captured through data-driven approaches. We aim to capture this relation utilizing a small number of latent variables that be used as both a low-dimensional representation of a vehicle geometry as well as the estimated drag coefficient. For this objective, we use an observable-augmented autoencoder assisted by principle component analysis (PCA)[32–36] that reconstructs the high-dimensional car vehicle information while estimating the vehicle drag coefficient from the latent space, as illustrated in Fig. 3.

**Latent manifold discovery**

If we directly attempt to learn a relationship between the vehicle geometries and the drag coefficient for a given dataset, we can learn many possible models that provide similar levels of accuracy. This can make shape optimization difficult, because such models may prove less reliable when generalizing to produce new geometries. With this in mind, we seek low-dimensional coordinates that capture a relationship between input geometries and the drag coefficient, while reconstructing a geometry from the low-dimensional representation. Training both tasks simultaneously helps to mitigate overfitting as well as allows us to observe geometric similarities, which is helpful for identifying relevant features for design optimization.

The original vehicle geometry is first preprocessed to a voxelized representation. This represents the dimensionality reduction step in Fig. 1, as we effectively reduce our high-fidelity input from a mesh represented with a set of points connected by smooth, continuous segments to a regular grid with finite resolution. The voxelized geometries are then used for identifying a low-dimensional latent space. What this amounts to is learning a curvilinear coordinate system that parameterizes a manifold representation of our dataset. We learn such a manifold by using a nonlinear autoencoder, a type of unsupervised machine-learning model[37]. A basic autoencoder consists of an encoder, which reduces the dimension of input data into a lower-

dimensional latent space, and a decoder which reconstructs the input from the encoded representation.

For this work, we leverage an observable-augmented nonlinear autoencoder[32] that is simultaneously trained to estimate the drag coefficient from the compressed representation of the vehicle geometries, as illustrated in Fig. 3. Since the drag coefficient needs to be estimated from the latent variables, the identified latent representation holds a relationship between vehicle geometries and their aerodynamic performance in a low-order manner. In other words, the neural network weights are trained such that features relevant to estimations of the drag coefficient are captured. This can also reduce the problem of analyzing qualitative similarities of geometric trends in the original high-dimensional geometries to observing changes in salient features extracted in our low-dimensional space. Moreover, such an aerodynamically relevant manifold provides a desired direction to improve aerodynamic performance in a low-order manner. In other words, we can identify an optimal modification of the vehicle design with reduced computational cost. Further details of this approach are given in the Methods section.

Shown in Fig. 4 is the discovered three-dimensional latent space manifold, $(\xi_1, \xi_2, \xi_3)$, that is learned by the present autoencoder. We note that for our dataset, a three-dimensional latent space is enough to achieve sufficiently reasonable geometry reconstruction as well as accurate drag estimation, as we observed very little gain in accuracy without a substantial increase in the latent space dimension. Through the drag decoder, we can obtain an estimated drag coefficient corresponding to the geometry parameterized by any point in our three-dimensional latent space in Fig. 4. We also note that the geometry reconstruction from the decoder is qualitatively indistinguishable from the original input for a number of cases, depicted in Figs. 3 and 4. The average percent absolute error of estimated drag values lies within 2% of the reference value for the training, validation, and test sets. As seen in Fig. 4, each point in the low-dimensional space corresponds to a vehicle geometry. We also observe that vehicles cluster in the low-order space based on both geometric similarity and the estimated drag coefficient with distinct point clouds corresponding to the different car types (shown by the marker shape). Increased estimated drag approximately correlates with increasing $\xi_1$ and $\xi_3$ in the shown manifold.

The relative distances between latent points can be taken to be representative of the similarity between vehicle cases. For example, vehicles with very different geometries and aerodynamic performance, such as sedans and box cars, are placed far apart in the latent space. On the other hand, many of the hatchback geometries have an intermediate drag value, which is reflected by their placement in between the low-drag sedans and the high-drag SUV and box cars.

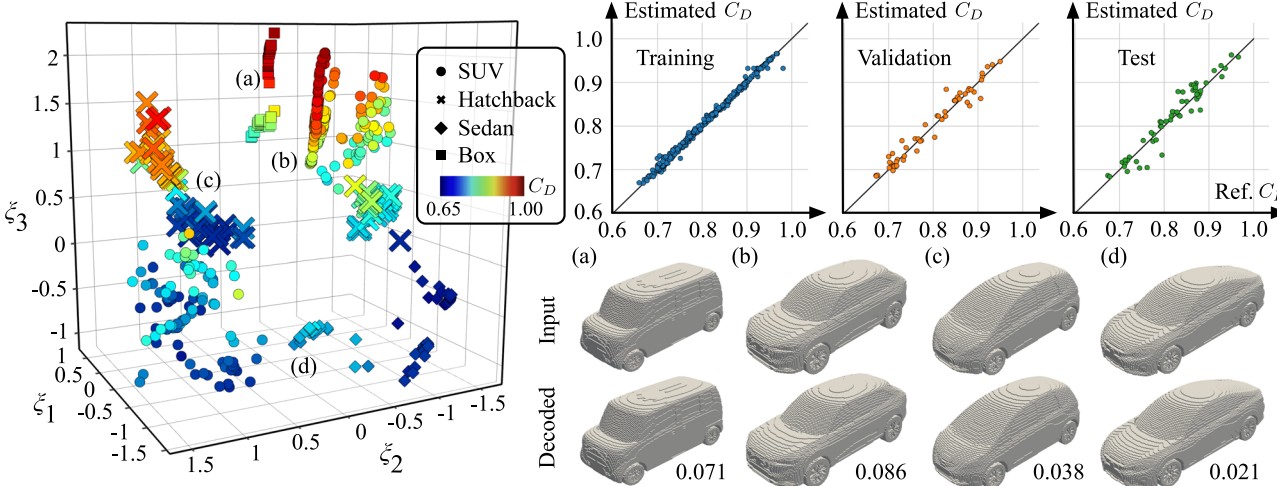

**Fig. 4 | Nonlinear machine-learning-based compression of vehicle geometries.** The left plot depicts the latent space of sport utility vehicles (SUVs), hatchbacks, sedans, and box geometries. Right parity plots show the comparison of estimated and reference normalized drag coefficients $C_D$. Comparisons of example vehicle geometries and autoencoder reconstructions are shown and the corresponding points marked in the latent space with **a–d**. The value underneath each decoded geometry reports the percent $L_2$ reconstruction error metric, $\varepsilon = \parallel \mathbf{p} - \hat{\mathbf{p}} \parallel_2 / \parallel \mathbf{p} \parallel_2$.

## Data-driven shape optimization

With the identified latent space manifold, we aim to modify existing vehicle geometries to improve aerodynamic performance. Since the present latent manifold relates car geometries with drag performance, optimization can be performed directly in the low-dimensional latent space rather than the high-dimensional space of the input data which can accelerate computations of the gradients. Knowing this, we obtain a direction in the latent space corresponding to a decrease in the drag coefficient for a given geometry by computing the gradient of drag at a given latent space point. By modifying the latent space coordinate in the direction of reduced drag and observing changes in the decoded geometry, we can identify regions of the vehicle geometry to modify in order to reduce the drag coefficient. The produced car geometries from this iterative descent process can then be used to generate baseline designs for further validation.

However, it is important that any modified geometries produced during this optimization process should not be "extrapolation" cases. That is to say, we must ensure that the geometry produced by our optimization process still resembles a physically realizable car design. If the optimized design is too dissimilar to any cases in training, this may also put the reliability of the estimated drag into question. Since the distance in the latent space can be considered a measure of similarity, the present data-driven optimization considers a constraint based on the latent space distance between the optimized design and the training data. We consider a soft distance constraint which penalizes the optimization from moving too far from the training data. This distance constraint adds a penalty term to the cost function, which grows as the distance from the training data increases. Further details of optimization formulation and the constraints are given in the Methods section.

We demonstrate our data-driven vehicle shape optimization on the discovered manifold in Fig. 5. We present an example case of geometry optimization starting from a high-drag SUV case (initial normalized drag coefficient, $C_D \approx 0.86$). The initial geometry and the geometry after performing the shape optimization with the soft constraint are also visualized in Fig. 5. The model estimates an 11% reduction of the normalized drag ($C_D \approx 0.77$) for the modified geometry. Along the optimization trajectory, we decode geometries and validate the drag estimate with LES performed on the smoothed voxel data. As seen in Fig. 5 we observe agreement between the trend of the estimated drag and the value corresponding to the CFD simulation for a few of the sampled validation cases.

In Fig. 6, we show the pressure fields around the validation geometry. In the optimized geometry, we observe that the rear geometry is modified.

The slope of the roof leading to the rear spoiler is lowered, and the end is shifted backward. This provides a boattail-like effect as this reduces the pressure gradient where the flow separates (at the spoiler), which in turn reduces the pressure drag. Additionally, there is a smaller observed pressure gradient around the C-pillar, and we note that the trailing wake vortices are elongated and tapered, reducing their influence on the induced drag. The edges of the front geometry are smoothed, which reduces the size of the front face, which the high-pressure zone (red) acts on, in addition to allowing the flow to smoothly transition around the front geometry. The validation cases demonstrate that the model has learned a trend between geometric features and aerodynamic performance rather than just memorizing the training data.

## Discussion

In the current study, we perform a data-driven analysis of the aerodynamic performance of production vehicle geometries, which have been obtained with experimentally validated LES. We utilize a data-driven vehicle shape optimization approach leveraging an observable-augmented nonlinear autoencoder which identifies a low-dimensional latent space manifold that provides a compressed representation of automobile geometries. The use of the observable augmentation enforces that the learned low-dimensional space also is relevant to the estimation of the drag coefficient. For the present dataset consisting of industry vehicle designs, the demonstrated approach effectively compresses voxelized geometries to just a three-dimensional latent space manifold while being able to sufficiently reconstruct the geometry and estimate the drag coefficient. This obtained low-dimensional space allows us to observe the relationship between different vehicle geometries and their respective aerodynamic performance, which we use to perform optimization of vehicle aerodynamics. We decode various geometries along the optimization trajectory and validate the trend of estimated drag with LES. Results of validation CFD for the decoded geometries show agreement with the trend of the estimated drag coefficients during the optimization process.

For the current autoencoder architecture, we employed a combination of PCA and a nonlinear multi-layer perceptron to improve the tractability and convergence of the neural network portion of the model by initially compressing the input voxel data with PCA. While we acknowledge the use of voxel geometry introduces cubic scaling of required storage memory with data resolution, we argue that the current approach of using voxel data with a PCA-assisted autoencoder offers a compromise between computational feasibility and geometric fidelity for use in an industrial setting, with the

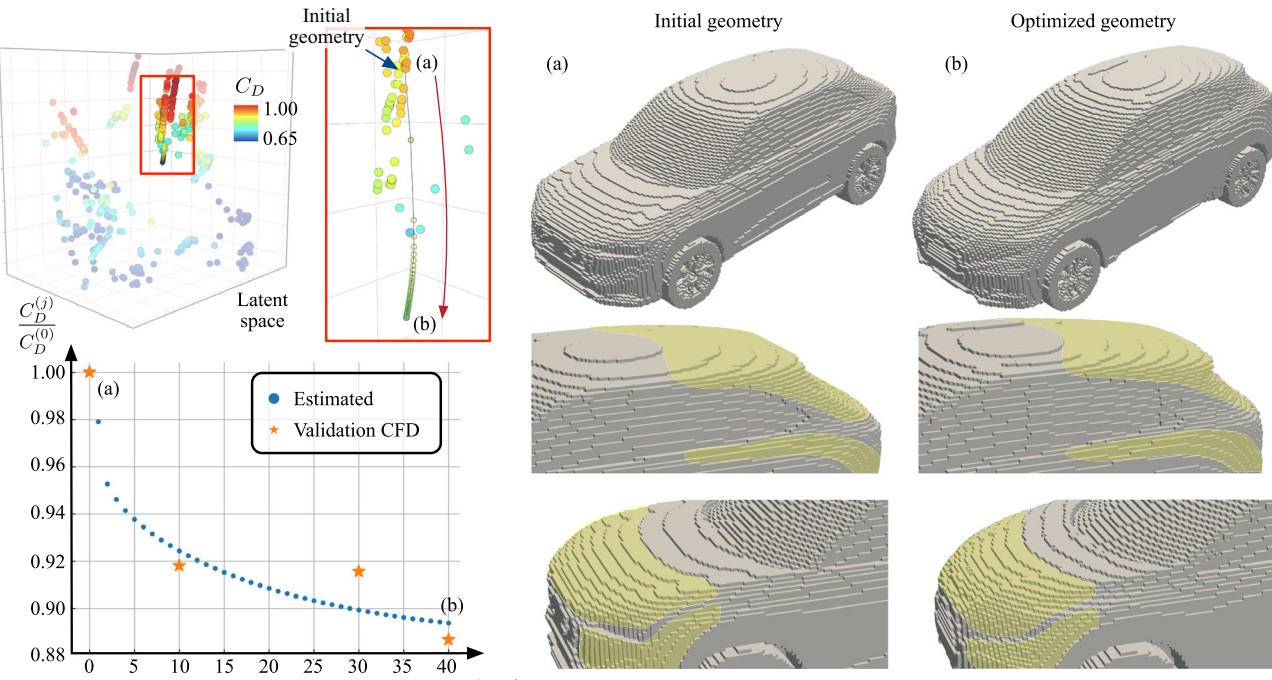

**Fig. 5 | Data-driven vehicle shape optimization in the discovered latent space colored by the normalized drag coefficient.** The latent optimization path with the corresponding normalized drag coefficient is shown. Orange points depict drag values obtained from validation computational fluid dynamics (CFD) simulation. **a** The given initial car and **b** optimized geometry are also depicted. The regions highlighted in yellow highlight some features that have been substantially modified.

present model taking only around a single hour to train on a standard laptop (compared to approximately $1.5 \times 10^4$ CPU hours for evaluating a single LES case). The choice to use voxel data instead of the original mesh geometry is because models based on mesh data often are limited to the deformation of existing control points and can produce distorted results for highly concave shapes or very sharp and thin features, especially when considering modifications like extrusion—limiting explorable shape modifications[24–26,38]. Another difficulty in mesh or point-based models for industrial use is that they typically require a fixed number of points at a relatively uniform density distribution, in addition to being computationally expensive during the graph construction process[39]. It is for these reasons we opt for a voxel-based representation. While the use of a PCA autoencoder approach makes the current approach computationally tractable with voxel data, it would be of interest to explore how alternative methods could be used to exploit local features to make complex geometric with minimal sacrifices of geometric information in a tractable manner.

The current study explores the potential avenue for a machine-learning analysis to expedite industrial aerodynamic design. We have demonstrated a proof-of-concept study of the applications of data-driven design to experimentally validate LES data of production vehicles for drag reduction. This analysis can also be extended to the optimization of different design parameters, such as other aerodynamic coefficients or relevant metrics like mass or volume. Machine learning has the capability to streamline the production of more efficient vehicles, which is especially beneficial to the transition to electric transportation. The use of data-driven methods as a tool to direct the iterative design process exhibits promise for accelerating industrial design optimization.

## Methods

### Automobile dataset preparation

For this study, we consider a dataset of over 500 geometries, which consists of a variety of industry car models generated from over 20 baseline car designs. The baseline car designs consist of SUVs, hatchbacks, sedans, and box cars of varying sizes and shapes taken from industrial geometries. For each baseline model, additional cases are generated by modifications of

various parameters such as the car width, height, and length. The flow fields around each vehicle are numerically simulated with a large-eddy simulation using a Helmholtz flow solver with a Vreman sub-grid model[13]. The free stream is specified with velocity $U_\infty = 140\ kmh^{-1}$, density $\rho_\infty = 1.205\ \mathrm{kgm}^{-3}$, and dynamic viscosity $\mu_\infty = 1.822 \times 10^{-5}\ m^2 s^{-1}$, resulting in a Reynolds number of up to around $1.18 \times 10^7$ with the car length as the characteristic length scale. The flow is computed with time step $\Delta t = 8 \times 10^{-5}\ s$ and is allowed to initialize for 0.32 s (4000 steps), with statistics averaged over the next 0.8 s (10,000 steps). Simulations are performed with the Fidelity CharLES flow solver[40]. The domain boundary conditions are chosen to emulate the experimental wind-tunnel setting. The floor geometry consists of no-slip walls and slip walls to represent the suction and non-suction regions of the wind-tunnel floor, and a moving ground condition is specified for the belt. The vehicle body is set to a wall boundary condition, and the tire and wheel geometries are set to a rotational velocity wall condition. Additionally, a moving mesh solver is used to compute the flow around the rotating wheel geometry[13]. Approximately, 64 million control volumes are contained in the mesh, with around 0.6 million for the rotating wheel geometry. The total computational time for a single case is around 14950 CPU hours. This simulation setup has been validated with wind-tunnel experiments to verify its accuracy in real-world conditions[13]. The drag coefficient is defined as $C_D = 2F_D/(\rho_\infty U_\infty^2 A)$ where $F_D$ is the drag force, and $A$ is the cross-sectional area. The pressure coefficient is defined as $C_p = 2(p - p_\infty)/(\rho_\infty U_\infty^2)$ where $p - p_\infty$ is the local static pressure. The total pressure coefficient is $C_{p,t} = 2p_t/(\rho_\infty U_\infty^2)$ where $p_t = p + \frac{1}{2}\rho U^2$ is the total pressure. All coefficients present in the current work have been additionally normalized to preserve confidentiality. The vehicle geometry is converted to a uniform voxel grid with a fixed resolution of 20 mm per voxel.

### PCA-assisted autoencoder setup

To identify a low-order manifold that captures the relationship between vehicle geometries and drag, we use an observable-augmented autoencoder consisting of a combination of principal component analysis and a multi-layer perceptron[32,35,37,41]. As illustrated in Fig. 3, the model consists of an encoder, a decoder, and a secondary drag decoder. The encoder,

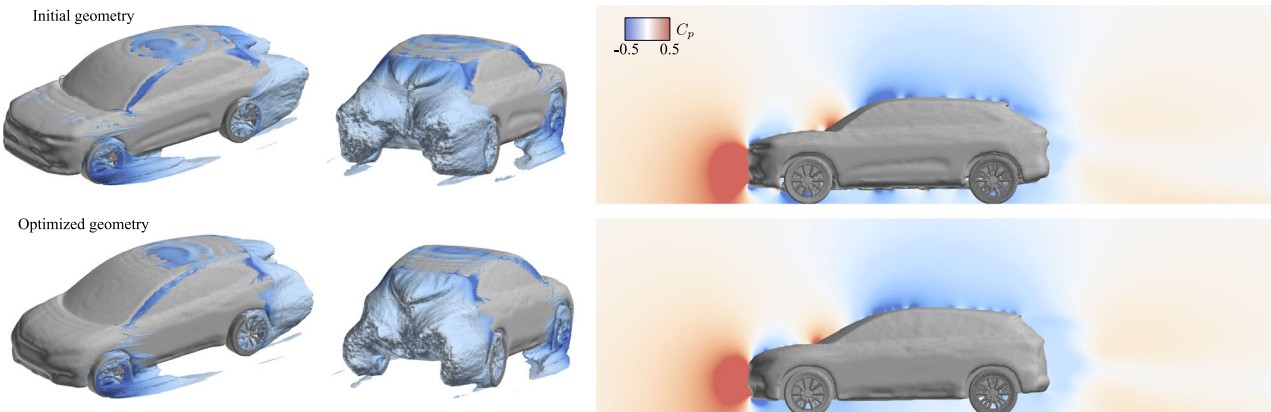

**Fig. 6 | Validation large-eddy simulation (LES) flow fields for smoothed voxel geometry for the initial high-drag SUV case and the modified geometry obtained from the optimization procedure.** (Left) Zero total pressure ($C_{p,t} = 0$) isocontour colored by the pressure coefficient, $C_p$. (Right) Midplane slice of pressure coefficient field, $C_p$. The pressure difference around the rear spoiler region is reduced.

$\mathcal{E} : \mathbb{R}^N \to \mathbb{R}^n$, takes the voxelized geometry as input $\mathbf{p} \in \mathbb{R}^N$ and produces a low-order representation of the geometry $\boldsymbol{\xi} \in \mathbb{R}^n$ with dimension $n < N$. On the other hand, the decoder, $\mathcal{D} : \mathbb{R}^n \to \mathbb{R}^N$, takes the low-order state $\boldsymbol{\xi}$ as input and produces a reconstruction of the input geometry, $\hat{\mathbf{p}} \in \mathbb{R}^N$. In addition, the secondary drag decoder, $\mathcal{F} : \mathbb{R}^n \to \mathbb{R}$, estimates the drag coefficient $C_D \in \mathbb{R}$ from the low-order representation[32].

We first perform principle component analysis (PCA)[33] of the voxelized body data to reduce the dimension of the geometry. The principle components give a set of orthogonal basis vectors, or modes, which capture variations in the shape designs with respect to the mean geometry. For a given dataset of geometries, $\{\mathbf{p}_i\}_{i=1}^M$, the PCA modes are given by the following optimization problem, which minimizes the difference between the original data and its projection onto a set of orthogonal PCA modes,

$$\{\boldsymbol{\phi}_k\}_{k=1}^r = \underset{\{\tilde{\boldsymbol{\phi}}_k\}_{k=1}^r}{\mathrm{argmin}} \sum_{i=1}^M \left\| \tilde{\mathbf{p}}_i - \sum_{k=1}^r a_{i,k} \tilde{\boldsymbol{\phi}}_k \right\|_2^2, \tag{1}$$

where $\{\boldsymbol{\phi}_k\}_{k=1}^r$ is the set of PCA modes, $\boldsymbol{\phi}_k \in \mathbb{R}^N$, and $\tilde{\mathbf{p}}_i = \mathbf{p}_i - \bar{\mathbf{p}}$ is the $i$th data input with the mean of the dataset removed. We have that the PCA coefficient, $a_{i,k} = \boldsymbol{\phi}_k^T \tilde{\mathbf{p}}_i \in \mathbb{R}$, is the inner product of the $i$th data vector and the $k$th PCA mode. The solutions to this optimization problem correspond to the eigenvectors of the covariance matrix. Given the PCA coefficients associated with an input, $\mathbf{p}_i$, we can obtain the reconstructed output with

$$\hat{\mathbf{p}}_i = \bar{\mathbf{p}} + \sum_{k=1}^r a_{i,k} \boldsymbol{\phi}_k = \bar{\mathbf{p}} + \boldsymbol{\Phi} \boldsymbol{a}_i, \tag{2}$$

where $\boldsymbol{\Phi}$ is a matrix consisting of the PCA modes $\{\boldsymbol{\phi}_k\}_{k=1}^r$ as columns and $\boldsymbol{a}_i$ is a column vector of the PCA coefficients.

The neural network portion of the autoencoder is a multi-layer perceptron (MLP)[41] which inputs the PCA coefficients of a vehicle geometry and outputs a reconstruction of the PCA coefficients and an estimate of the drag. The MLP finds a low-dimensional embedding of the PCA coefficients, $\boldsymbol{\xi}$, which it uses to produce a reconstruction of the PCA coefficients as well as the drag estimate. The output PCA coefficients are then used to obtain a reconstruction of the original input data using Eq. (2). In this sense, it can be seen as a nonlinear correction to the linear decomposition of PCA[35]. The nonlinear activation functions that we use for the MLP layers are hyperbolic tangent, $\tanh(s) = (e^s - e^{-s})/(e^s + e^{-s})$. The network weights, $\mathbf{w}$, are obtained by minimizing a weighted sum of the mean squared error of the

PCA coefficient reconstruction and the drag coefficient estimation:

$$\mathbf{w}^* = \underset{\mathbf{w}}{\mathrm{argmin}} \left[ \frac{1}{M} \sum_{i=1}^M \left( ||\boldsymbol{a}_{\mathrm{in,i}} - \mathcal{D}(\mathcal{E}(\boldsymbol{a}_{\mathrm{in,i}}; \mathbf{w}_{\mathcal{E}}); \mathbf{w}_{\mathcal{D}})||_2^2 \right. \right.$$
$$\left. \left. + \beta ||C_{D,\mathrm{ref},i} - \mathcal{F}(\mathcal{E}(\boldsymbol{a}_{\mathrm{in,i}}; \mathbf{w}_{\mathcal{E}}); \mathbf{w}_{\mathcal{F}})||_2^2 \right) \right], \tag{3}$$

where $\beta$ is a weighting parameter that adjusts the relative importance of the error terms associated with the $C_D$ estimation and the PCA coefficient estimation. This parameter was chosen based on an L-curve analysis[42] and was taken to be $\beta = 1 \times 10^5$ for our current model. The present MLP encoder takes in 400 PCA coefficients as input and consists of 8 layers with node counts of 400-512-256-128-64-32-16-3 with the MLP decoder symmetrically constructed. The drag decoder consists of 7 layers with the node counts of 3-16-32-64-32-16-1. We split our dataset into training, validation, and test sets with a random 80:10:10 split. We use the Adam optimizer[43] for 1000 epochs with an early stopping criterion to avoid overfitting[44]. During training, we save the model parameters when the loss for the validation set improves.

## Shape optimization procedure

With the trained autoencoder model, we can perform geometry optimization to reduce the drag of a given design in an iterative manner. Consider a geometry at iteration $j$ which we denote $\mathbf{p}^{(j)}$. We prepare the corresponding PCA coefficients, $\boldsymbol{a}^{(j)} = \boldsymbol{\Phi}^T \mathbf{p}^{(j)}$, and encode them to find the corresponding latent point $\boldsymbol{\xi}^{(j)} = \mathcal{E}(\boldsymbol{a}^{(j)})$, as well as the corresponding drag estimate $C_D^{(j)} = \mathcal{F}(\boldsymbol{\xi}^{(j)})$. By backpropagating through the drag decoder, we can obtain the sensitivity of the drag estimate with respect to the latent space point to obtain the direction of decreasing drag in the latent space, $\mathbf{d}^{(j)} = -\nabla_{\boldsymbol{\xi}} C_D|_{\boldsymbol{\xi} = \boldsymbol{\xi}^{(j)}}$[45]. With a user-specified step size, $h^{(j)}$, the modification in the latent space with respect to decreasing drag is $\Delta \boldsymbol{\xi}^{(j)} = h^{(j)} \mathbf{d}^{(j)}$.

To achieve a reliable geometry reconstruction and drag coefficient estimation, we also consider a penalty such that we do not move too far away from the training data. In the present work, we choose a threshold distance $d_{\mathrm{thresh}}$, to be the average value of the maximum distances between the training points. At every iteration, the distance between the latent space point, $\boldsymbol{\xi}^{(j)}$, and the nearest training data point in the latent space $\boldsymbol{\xi}_{\mathrm{train}}$ is measured. We modify the latent space perturbation to be the following:

$$\Delta \tilde{\boldsymbol{\xi}}^{(j)} = h^{(j)} \left[ \mathbf{d}^{(j)} - \alpha \frac{(\boldsymbol{\xi}^{(j)} - \boldsymbol{\xi}_{\mathrm{train}})}{d_{\mathrm{thresh}}^2} \left( 1 - \exp\left( - \| \boldsymbol{\xi}^{(j)} - \boldsymbol{\xi}_{\mathrm{train}} \|_2^2 / d_{\mathrm{thresh}}^2 \right) \right) \right]. \tag{4}$$

The first term corresponds to the direction of decreasing drag in the latent space, while the second term penalizes moving away from the training data. The term $\alpha$ controls the influence of the soft penalty term.

Once we obtain a perturbation in the latent space, we derive a new latent point, $\xi^{(j+1)} = \xi^{(j)} + \Delta\tilde{\xi}^{(j)}$. Using the decoder, the corresponding geometry $\mathbf{p}^{(j+1)} = \mathcal{D}(\xi^{(j+1)})$ and drag estimate $C_D^{(j+1)} = \mathcal{F}(\xi^{(j+1)})$ can be computed. For the proof of concept shown in this work, we perform this iteration starting from a high-drag SUV geometry until the percent change of the estimated $C_D$ drops under $10^{-3}$ (which resulted in 40 iterations) with $h = 0.02$ and $\alpha = 0.125$.

## Data availability
The data that support the findings of this study are available from the corresponding author upon reasonable request.

## Code availability
The codes that support the findings of this study are available from the corresponding author upon reasonable request.

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

## Acknowledgements

J.T., K.F., and K.T. thank Honda Motor Co., Ltd. for supporting this research.

## Author contributions

J.T., K.F., and K.I. developed the software and visualized the results. J.T., K.F., and K.T. conceptualized the approach and wrote the manuscript. K.I., D.U., and Y.O. performed numerical simulations for dataset preparation and validation. K.T. and K.O. secured funding support and supervised the project.

## Competing interests

The authors declare no competing interests.
