## [Peer Review file · Communications Engineering]

Aerodynamics-guided machine learning for design optimization of electric vehicles

Corresponding Author: Professor Kunihiko Taira

Version 0:

Reviewer comments:

Reviewer #1

(Remarks to the Author)

Coming from a perspective of primarily a learning researcher, I am not sure I can fully appreciate the contribution of this paper. The learning method used in the paper appears to be off-the-shelf manifold learning, and the (hyper)parameters used appear to be devised by trial and error (e.g., for the latent space dimension, "we note very little gain in accuracy without a substantial increase in the latent space dimension").

I am not making these observations as a complaint, just as a rationale for my conclusion that the true contribution of this paper can only lie in its application to real data and subsequent shape optimization. On those points, I believe the authors' results (although, as stated, they is far from my field of expertise), but I am left wondering about benchmarks: the paper makes an argument that explicitly data-driven methods (i.e., learning) can be better and faster than traditional analysis. It would be good to offer some -- inasmuch as possible, quantified -- support to that claim. While there is a written discussion in the shape optimization section, I am wondering if it could be explicitly said what the insights are that would not have been easily obtainable using traditional methods, why they would not be obtainable using traditional methods, and how much they affect performance.

Reviewer #2

(Remarks to the Author)

The reviewer appreciates the work and effort put in by the authors. However, it is tough to justify the novelty of this work, especially, for a journal publication.

A non-comprehensive list of previous work in this space below:

There are has been extensive research in using data-driven surrogate models for drag prediction. Few examples below:

Learning mesh-based simulation with graph networks.

Prediction of Aerodynamic Flow Fields Using Convolutional Neural Networks

Geometry-Informed Neural Operator for Large-Scale 3D PDEs

DrivAerNet: A Parametric Car Dataset for Data-Driven Aerodynamic Design and Graph-Based Drag Prediction

Several papers have used the concept of latent space:

Concept Activation Vectors for Generating User-Defined 3D Shapes

Surrogate Modeling of Car Drag Coefficient with Depth and Normal Renderings

On the efficiency of a point cloud autoencoder as a geometric representation for shape optimization

Point2ffd: Learning shape representations of simulation-ready 3d models for engineering design optimization

Exploiting local geometric features in vehicle design optimization with 3d point cloud autoencoders

Scalability of learning tasks on 3d cae models using point cloud autoencoders

Several papers have discussed optimization techniques:

A Data-Driven Interactive System for Aerodynamic and User-centred Generative Vehicle Design

Machine learning in aerodynamic shape optimization

Multi-fidelity deep learning for aerodynamic shape optimization using convolutional neural network

This paper - Concept Activation Vectors for Generating User-Defined 3D Shapes - especially, is very similar to to the current paper.

Given the extensive body of published work, it is difficult to justify the novelty of this paper for an archival journal publication.

Reviewer #3

(Remarks to the Author)

This paper proposes a data-driven strategy that combines a nonlinear autoencoder and high-fidelity simulation data to optimize the aerodynamic design of electric vehicles. I have below the comments to the authors.

1. The introduction effectively outlines the context and importance of aerodynamic design in electric vehicles. However, the transition from traditional methods to the proposed data-driven approach could be elaborated further. Clarifying how the proposed method specifically improves over traditional aerodynamic optimization techniques will help in understanding the novel contributions of your work more distinctly.
2. The methodology section describes the use of a nonlinear autoencoder for deriving low-dimensional representations of vehicle geometries and their aerodynamic performance. It would enhance the paper if the authors could provide more detailed justification for choosing this specific machine learning model over others. Additionally, a comparison with other potential models could provide insight into the robustness and efficiency of the chosen method.
3. The results section demonstrates the capability of the proposed method in optimizing vehicle shapes for better aerodynamic performance. However, the validation of these results appears limited to comparisons within the dataset itself. It would be beneficial if external validation or cross-validation methods were applied to ensure that the model's predictions are robust and applicable in real-world scenarios. This could involve testing the optimized designs in physical wind tunnel experiments or through independent computational fluid dynamics simulations.
4. While the manuscript touches upon the computational advantages of the proposed method, a detailed analysis comparing the computational costs (both time and resources) against traditional methods is missing. Providing explicit metrics on computational savings could significantly strengthen the case for the proposed data-driven approach.
5. The discussion section could be expanded to include a more detailed examination of the limitations of the current study. For instance, the impact of varying mesh resolutions on the model's accuracy or the sensitivity of the model to different types of vehicle geometries could be explored. Additionally, outlining potential future work, such as integrating other vehicle performance metrics beyond aerodynamics (like weight or electric consumption), would provide a roadmap for extending the research.
6. More details on the experimental setup and data collection process used to obtain the vehicle geometries and their aerodynamic performance would enhance the reliability of the study. This includes specifying the conditions under which the data were collected, any preprocessing steps taken, and the criteria used for data inclusion or exclusion.

Version 1:

Reviewer comments:

Reviewer #1

(Remarks to the Author)

I thank the authors for the clarifications relevant to my comments. As stated previously, the nature of the paper's claimed contribution -- aerodynamic analysis and design -- essentially excludes my expertise, which is in learning. I appreciate the short discussion on the benefits from the standpoint of computational complexity, although I am pointing out that the big-O notation in "O(10⁴) hours" is misused: big-O has to do with asymptotic analysis, not with order of magnitude or approximate amounts.

Altogether, since the authors and I agree that the paper's claimed contribution is not in novel learning methods, I have no further comments and defer to more qualified reviewers to evaluate the application content.

Reviewer #2

(Remarks to the Author)

Reviewer #3

(Remarks to the Author)

The authors addressed my previous comments and I don't have comments further

Response to Reviewer 1

Journal: Communications Engineering

Title: Aerodynamics-guided machine learning for design optimization of electric vehicles

Authors: Jonathan Tran, Kai Fukami, Kenta Inada, Daisuke Umehara, Yoshimichi Ono, Kenta Ogawa, and Kunihiko Taira

Date sent: August 26, 2024

The authors would like to thank the reviewers for taking their time to review our manuscript. Based on the reviewers' comments, revisions have been made to the manuscript, with major changes highlighted in red. We hope our responses fully address the comments from the reviewer. In what follows, we list the reviewer's comments with our responses shown in blue.

1. Coming from a perspective of primarily a learning researcher, I am not sure I can fully appreciate the contribution of this paper. The learning method used in the paper appears to be off-the-shelf manifold learning, and the (hyper)parameters used appear to be devised by trial and error (e.g., for the latent space dimension, "we note very little gain in accuracy without a substantial increase in the latent space dimension").

Response: We agree with the reviewer's initial point – that is, from a machine learning perspective we are making use of existing manifold learning and optimization procedures. However, we would like to note that the novelty of the current work lies not only in the application to real industry designs but in the generation and aerodynamic analysis of the dataset itself. In comparison to the most comprehensive existing studies (e.g. DrivAerNet, Elrefaie et al., 2024) we use significantly higher-fidelity simulations that more accurately capture aerodynamic phenomena for real vehicles. The current work utilizes large-eddy simulations (LES) with production vehicle designs for the first time. The computational setup more accurately reflects the conditions of the physical wind tunnel testing environment which was used to validate the simulation (Nagaoka et al., 2024). Previous studies make use of a steady-state Reynolds-averaged Navier–Stokes (RANS) equations solver at lower fidelity and velocities which results in less accurate results than LES. It should be made clear that the difference is not simply a matter of numerical accuracy, but it is that entirely different physical phenomena are captured with our more accurate simulations, such as the existence of a wheel wake from the rotating mesh solver. We have changed the discussion in the introduction and analysis (pages 3,4) as well as the discussion of the simulations (page 8) to more directly address these details. We also refer readers to the cited source for further discussion on the simulations.

- Elrefaie, M., Morar, F., Dai, A., & Ahmed, F. (2024). DrivAerNet++: A Large-Scale Multimodal Car Dataset with Computational Fluid Dynamics Simulations and Deep Learning Benchmarks. arXiv preprint arXiv:2406.09624.
- Nagaoka, H., Yenerdag, B., Ambo, K., Philips, D., Ivey, C., Brès, G., & Bose, S. (2024). Prediction of aerodynamic drag in SUVs with different specifications by using Large-Eddy simulations. SAE Technical Paper, No. 2024-01-2525.

2. I am not making these observations as a complaint, just as a rationale for my conclusion that the true contribution of this paper can only lie in its application to real data and subsequent shape optimization. On those points, I believe the authors' results (although, as stated, they is far from my field of expertise), but I am left wondering about benchmarks: the paper makes an argument that explicitly data-driven methods (i.e., learning) can be better and faster than traditional analysis. It would be good to offer some – inasmuch as possible, quantified – support to that claim. While there is a written discussion in the shape optimization section, I am wondering if it could be explicitly said what the insights are that would not have been easily obtainable using traditional methods, why they would not be obtainable using traditional methods, and how much they affect performance.

Response: The reviewer asks for substantiation of how data-driven analysis can outperform traditional analysis, which is a good point to be made. We make these arguments from a heuristic perspective. Traditional methods would require an aerodynamicist to parse highly complex, nonlinear fluid interactions from a vast amount of trials to understand how one may improve aerodynamic performance with specific modifications. Data-driven methods can assist (not entirely replace) such analysis by producing plausible modifications that should be then validated with traditional means. The time to run a single LES case at a satisfactory resolution for industry use can take up to 14,950 CPU hours on a computing cluster. Compared to the estimated training and evaluation time of the data-driven approach which is on the order of a single hour, and can be run on a consumer laptop, we argue that data-driven analysis can result in significant time savings. We update our discussion on pages 7 and 8 to reference the time costs.

Response to Reviewer 2

Journal: Communications Engineering

Title: Aerodynamics-guided machine learning for design optimization of electric vehicles

Authors: Jonathan Tran, Kai Fukami, Kenta Inada, Daisuke Umehara, Yoshimichi Ono, Kenta Ogawa, and Kunihiko Taira

Date sent: August 26, 2024

The authors would like to thank the reviewers for taking their time to review our manuscript. Based on the reviewers' comments, revisions have been made to the manuscript, with major changes highlighted in **red**. We hope our responses fully address the comments from the reviewer. In what follows, we list the reviewer's comments with our responses shown in **blue**.

1. The reviewer appreciates the work and effort put in by the authors. However, it is tough to justify the novelty of this work, especially, for a journal publication.

A non-comprehensive list of previous work in this space below:

There are has been extensive research in using data-driven surrogate models for drag prediction. Few examples below:

- Learning mesh-based simulation with graph networks.
- Prediction of Aerodynamic Flow Fields Using Convolutional Neural Networks
- Geometry-Informed Neural Operator for Large-Scale 3D PDEs
- DrivAerNet: A Parametric Car Dataset for Data-Driven Aerodynamic Design and Graph-Based Drag Prediction

Response: We agree with the reviewer's point that there are studies in data-driven surrogate models for drag prediction. From a machine-learning perspective, we have opted for possibly one of the most simple approaches. However, we note that the novelty of the current work lies in the application to real industry designs as well as in the generation and aerodynamic analysis of the dataset itself, which we argue is substantially more suitable for industrial use.

It is to our knowledge that in comparison to the most comprehensive existing studies (e.g. DrivAerNet++, Elrefaie et al., 2024) we use significantly higher-fidelity simulations that more accurately capture aerodynamic phenomena for real vehicles. The current work utilizes large-eddy simulations (LES) with a moving mesh solver and production vehicle designs. Additionally, the computational setup has been experimentally validated with Honda's 5 belt wind tunnel testing environment (Nagaoka et al., 2024), which to our knowledge has not been performed in previous works.

Previous studies make use of steady-state Reynolds-averaged Navier–Stokes (RANS) equations solver at lower fidelity and velocities, typically with less complex vehicle geometry which results in less accurate results than LES (Elrefaie et al., 2024). Previous studies have not performed simulations at a similar complexity in terms of computational setup and simulation conditions. It should be made clear that the difference is not simply a matter of numerical accuracy of the drag estimate, but it is that entirely different physical phenomena are captured with our more accurate simulations. One example is the wake generated from outflow from the rotating front wheel geometry which is not captured without the moving mesh solver, even with the use of LES (Nagaoka et al., 2024).

We have changed the discussion in the introduction and results (pages 3,4) and the discourse of the simulation (page 8) to focus more specifically on these details. However, we have omitted a more detailed discussion of the computational setup in favor of a discussion of the applications. Instead, we refer readers to the referenced sources.

- Elrefaie, M., Morar, F., Dai, A., & Ahmed, F. (2024). DrivAerNet++: A Large-Scale Multimodal Car Dataset with Computational Fluid Dynamics Simulations and Deep Learning Benchmarks. arXiv preprint arXiv:2406.09624.
- Nagaoka, H., Yenerdag, B., Ambo, K., Philips, D., Ivey, C., Brès, G., & Bose, S. (2024). Prediction of aerodynamic drag in SUVs with different specifications by using Large-Eddy simulations. SAE Technical Paper, No. 2024-01-2525.

2. Several papers have used the concept of latent space:

- Concept Activation Vectors for Generating User-Defined 3D Shapes.
- Surrogate Modeling of Car Drag Coefficient with Depth and Normal Renderings
- On the efficiency of a point cloud autoencoder as a geometric representation for shape optimization

- Point2ffd: Learning shape representations of simulation-ready 3d models for engineering design optimization
- Exploiting local geometric features in vehicle design optimization with 3d point cloud autoencoders
- Scalability of learning tasks on 3d cae models using point cloud autoencoders

Several papers have discussed optimization techniques:

- A Data-Driven Interactive System for Aerodynamic and User-centred Generative Vehicle Design
- Machine learning in aerodynamic shape optimization
- Multi-fidelity deep learning for aerodynamic shape optimization using convolutional neural network

This paper - Concept Activation Vectors for Generating User-Defined 3D Shapes - especially, is very similar to to the current paper.

Given the extensive body of published work, it is difficult to justify the novelty of this paper for an archival journal publication.

Response: We agree that vehicle shape optimization has been discussed previously, with much more elaborate approaches. However, the novelty of the present study is that it presents a more realistic application of data-driven analysis to an industrial setting for a few reasons. Beyond what has already been mentioned about the quality of the data set, none of the previous studies validate the results of the shape optimization or decoded geometries (outside of the test data) with any aerodynamic analysis (much less that which has been validated by industrial wind tunnel experiments). We perform an LES on geometries outside that of our validation and test dataset to verify the accuracy of the learned latent trends.

In the present study, not only do we validate various decoded geometries (a few of which are shown in the results), but we achieve satisfactory results on highly complex production designs with a simple model that can be run on a consumer laptop. The time to run a single LES case at a satisfactory resolution for industry use can take up to 14,950 CPU hours on a computing cluster. Compared to the estimated training and evaluation time of this simple data-driven approach is on the order of a single hour, we observe significant time savings. Our discussions on pages 7 and 8 are updated to reflect these comments

Response to Reviewer 3

Journal: Communications Engineering

Title: Aerodynamics-guided machine learning for design optimization of electric vehicles

Authors: Jonathan Tran, Kai Fukami, Kenta Inada, Daisuke Umehara, Yoshimichi Ono, Kenta Ogawa, and Kunihiko Taira

Date sent: August 26, 2024

The authors would like to thank the reviewers for taking their time to review our manuscript. Based on the reviewers' comments, revisions have been made to the manuscript, with major changes highlighted in red. We hope our responses fully address the comments from the reviewer. In what follows, we list the reviewer's comments with our responses shown in blue.

.....

1. The introduction effectively outlines the context and importance of aerodynamic design in electric vehicles. However, the transition from traditional methods to the proposed data-driven approach could be elaborated further. Clarifying how the proposed method specifically improves over traditional aerodynamic optimization techniques will help in understanding the novel contributions of your work more distinctly.

Response: Thank you for the helpful suggestion. Traditional methods would require an aerodynamicist to parse highly complex, nonlinear fluid interactions from a vast amount of trials to understand how one may improve aerodynamic performance with specific modifications. Data-driven methods can assist (not entirely replace) such analysis by producing plausible modifications that should be then validated with traditional means. We update our discussion on the introduction and on pages 7 and 8 to more directly address these comments.

2. The methodology section describes the use of a nonlinear autoencoder for deriving low-dimensional representations of vehicle geometries and their aerodynamic performance. It would enhance the paper if the authors could provide more detailed justification for choosing this specific machine learning model over others. Additionally, a comparison with other potential models could provide insight into the robustness and efficiency of the chosen method.

Response: We justify the choice of our model for a few reasons. Models based on graph or point cloud data often employ methods to discretize meshes into a suitable representation. Existing mesh-based methods include the use of "shrink-wrapping" or "polycubes" which can produce distorted results for highly concave shapes or very sharp and thin features (Umetani et al., 2018, Rios et al., 2021). This can produce very poor representations of important features for aerodynamic prediction, such as underbody geometry. Additionally, such mesh based methods consider only warping of a fixed number of control points, making exploration of more complex geometric modifications, such as extrusions, difficult. It is for these reasons that we opt for a voxel representation rather than a point/graph based model.

As for the specific choice of the PCA based autoencoder, the current approach of using PCA and an MLP offers a compromise between computational feasibility and geometric fidelity for use in an industrial setting, with the present model taking only around a single hour to train on a consumer-grade laptop. It has been previously found that a simple autoencoder and the combination of PCA + nonlinear autoencoder performed similarly for shape generation and interpolation with the major difference being the usage of local features (Rios et al., 2021). Knowing this, we choose the PCA approach as the model training does not require the full state input (which is already on the order of 2 million voxels for a 20 mm resolution). We do note that the present work can be easily applied to a fully convolutional model if computational resources permit. We update our discussion on pages 7 and 8 to address the reviewer's comments.

- Umetani, N., & Bickel, B. (2018). Learning three-dimensional flow for interactive aerodynamic design, ACM Transactions on Graphics, 37(4), 1-10.
- Rios, T., van Stein, B., Wollstadt, P., Bäck, T., Sendhoff, B., & Menzel, S. (2021). Exploiting local geometric features in vehicle design optimization with 3D point cloud autoencoders, IEEE Congress on Evolutionary Computation, 514-521.

3. The results section demonstrates the capability of the proposed method in optimizing vehicle shapes for better aerodynamic performance. However, the validation of these results appears limited to comparisons within the dataset

itself. It would be beneficial if external validation or cross-validation methods were applied to ensure that the model's predictions are robust and applicable in real-world scenarios. This could involve testing the optimized designs in physical wind tunnel experiments or through independent computational fluid dynamics simulations.

Response: The reviewer has made a good point referring to the validation of the current model. We want to note that the validation was performed in a few ways.

The computational setup for the Large Eddy Simulations was validated with wind-tunnel experiments at Honda's 5 belt wind tunnel facility (Nagaoka et al., 2024). We have added a note of this in the discussion of the simulations on page 8.

As for the validation of the aerodynamic predictions of the machine learning model, we first performed a cross-validation study on withheld cases with a train-validation-test split ratio of 80-10-10. Additionally, we performed an independent CFD study of generated designs during the shape optimization process which are not in the dataset as shown in Figures 5 and 6, using the same computational setup that has been experimentally validated. It is to our knowledge that existing studies have not performed validation analyses to the extent that we have. We have updated the discussion on page 7 to make these points more clear.

- Nagaoka, H., Yenerdag, B., Ambo, K., Philips, D., Ivey, C., Brès, G., & Bose, S. (2024). Prediction of aerodynamic drag in SUVs with different specifications by using Large-Eddy simulations. SAE Technical Paper, No. 2024-01-2525.

4. While the manuscript touches upon the computational advantages of the proposed method, a detailed analysis comparing the computational costs (both time and resources) against traditional methods is missing. Providing explicit metrics on computational savings could significantly strengthen the case for the proposed data-driven approach.

Response: We agree that we could be more specific on explicit metrics in comparison to traditional analysis. The time to run a single LES case at a satisfactory resolution for industry use can take up to 14,950 CPU hours on a computing cluster. Compared to the estimated training and evaluation time of the data-driven approach which is on the order of a single hour, and can be run on a consumer laptop, we argue that data-driven analysis can result in significant time savings. We update our discussion on pages 7 and 8 to reference these comments.

5. The discussion section could be expanded to include a more detailed examination of the limitations of the current study. For instance, the impact of varying mesh resolutions on the model's accuracy or the sensitivity of the model to different types of vehicle geometries could be explored. Additionally, outlining potential future work, such as integrating other vehicle performance metrics beyond aerodynamics (like weight or electric consumption), would provide a roadmap for extending the research.

Response: We have updated the discussion on pages 7 and 8 to discuss current limitations and future directions for study. One limitation comes from choice of the voxelized data representation. While it may prove to be a more accurate representation of some features, the required memory scales cubically with the data resolution. Currently, we use PCA to make training and evaluating models computationally tractable. However, it would be of interest to explore different methods to parameterize vehicle geometries that can accurately capture local geometric features while retaining necessary fidelity for aerodynamic analysis. We also outline extensions of the work to additional relevant parameters. This includes aerodynamic coefficients, like downforce/lift or sideforce, which are important for stability and safety, or other relevant parameters like mass or volume.

6. More details on the experimental setup and data collection process used to obtain the vehicle geometries and their aerodynamic performance would enhance the reliability of the study. This includes specifying the conditions under which the data were collected, any preprocessing steps taken, and the criteria used for data inclusion or exclusion.

Response: We have updated the methods discussion on page 8 to include more details of the aerodynamic analysis. For the aerodynamic data collection, the flow is allowed to initialize for 0.32 s (4000 steps), with statistics averaged over the next 0.8 s (10000 steps) to compute the average flow without transient effects from the flow field initialization. Baseline geometries were taken from existing production designs and were parametrically modified. Due to the page limitation of Communications Engineering, we refer readers to more detailed discussion of the aerodynamic data generation.

- Nagaoka, H., Yenerdag, B., Ambo, K., Philips, D., Ivey, C., Brès, G., & Bose, S. (2024). Prediction of aerodynamic drag in SUVs with different specifications by using Large-Eddy simulations. SAE Technical Paper, No. 2024-01-2525.